# Advancements and Obstacles of PARP Inhibitors in Gastric Cancer

**DOI:** 10.3390/cancers15215114

**Published:** 2023-10-24

**Authors:** Hongjie Chen, Yangchan Hu, Zirui Zhuang, Dingyi Wang, Zu Ye, Ji Jing, Xiangdong Cheng

**Affiliations:** 1College of Pharmaceutical Science, Zhejiang University of Technology, Hangzhou 310014, China; 2112107305@zjut.edu.cn (H.C.); 211122070123@zjut.edu.cn (Y.H.); 2112107129@zjut.edu.cn (D.W.); 2Zhejiang Cancer Hospital, Hangzhou Institute of Medicine (HIM), Chinese Academy of Sciences, Hangzhou 310022, China; zhuangzirui22@mails.ucas.ac.cn; 3School of Molecular Medicine, Hangzhou Institute for Advanced Study, University of Chinese Academy of Sciences (UCAS), Hangzhou 310024, China; 4Shanghai Institute of Materia Medica, Chinese Academy of Sciences, Shanghai 201203, China; 5Department of Gastric Surgery, Zhejiang Cancer Hospital, Hangzhou Institute of Medicine (HIM), Chinese Academy of Sciences, Hangzhou 310022, China; yezuqscx@zju.edu.cn; 6Zhejiang Key Laboratory of Prevention, Diagnosis and Therapy of Upper Gastrointestinal Cancer, Hangzhou 310022, China; 7Zhejiang Provincial Research Center for Upper Gastrointestinal Tract Cancer, Zhejiang Cancer Hospital, Hangzhou 310022, China

**Keywords:** gastric cancer, DNA damage repair, PARP inhibitor, cancer therapy

## Abstract

**Simple Summary:**

Gastric cancer (GC) is a highly aggressive malignancy affecting the digestive system and characterized by notable variations in its nature. Although survival rates have observed some enhancements, the prognosis remains grim, and mortality rates remain high. There is an immediate demand for innovative treatment strategies. Recent investigations have identified frequent mutations in genes responsible for repairing DNA damage in GC patients, highlighting the potential of PARP inhibitors (PARPi) as a promising therapeutic option. This article offers a comprehensive examination of the progress and hurdles in the development of PARPi for treating GC.

**Abstract:**

Gastric cancer (GC) is a common and aggressive cancer of the digestive system, exhibiting high aggressiveness and significant heterogeneity. Despite advancements in improving survival rates over the past few decades, GC continues to carry a worrisome prognosis and notable mortality. As a result, there is an urgent need for novel therapeutic approaches to address GC. Recent targeted sequencing studies have revealed frequent mutations in DNA damage repair (DDR) pathway genes in many GC patients. These mutations lead to an increased reliance on poly (adenosine diphosphate-ribose) polymerase (PARP) for DNA repair, making PARP inhibitors (PARPi) a promising treatment option for GC. This article presents a comprehensive overview of the rationale and development of PARPi, highlighting its progress and challenges in both preclinical and clinical research for treating GC.

## 1. Introduction

Gastric cancer (GC) is a prevalent malignancy globally, with over one million new cases reported in 2020 and a staggering 768,793 deaths, ranking fifth in incidence and fourth in mortality worldwide [1,2,3]. Surgical resection currently serves as the primary treatment for GC [4]. However, early-stage GC often lacks symptoms, leading to low rates of diagnosis and a higher proportion of patients being detected at an advanced stage [5]. Consequently, the survival rate remains poor and recurrence rates remain high despite surgical intervention [6]. Although advancements in radiotherapy, chemotherapy, and neoadjuvant therapies have improved survival rates over the years, the benefits derived from these treatments are limited, and resistance occurs frequently [7,8]. Hence, the current outlook for GC treatment is not optimistic, emphasizing the urgent need for novel therapeutic approaches.

As we are well aware, cancer is characterized by genomic instability, where DNA damage accumulates and remains unrepaired; this is known as DNA damage repair (DDR) deficiency [9]. These defects in DDR can lead to genetic alterations in cancer cells, including rearrangements, gene copy number changes, and mutations. Interestingly, these defects also create vulnerabilities that are specific to cancer cells, making them potential targets for cancer therapy [10]. Thus, there is a significant scientific interest in investigating therapeutic approaches that target the DDR process [11]. PARP inhibitors (PARPi), in particular, have gained significant attention as a promising avenue for cancer treatment, especially in cancer cells with breast cancer susceptibility gene (BRCA) mutations. This has opened up new avenues of research, exploring biomarker-directed “synthetic lethal” therapeutic strategies for various types of cancer [12,13]. An extensive investigation encompassing 2019 samples of GC from 12 publicly available datasets, including data from Harbin Medical University Cancer Hospital and The Cancer Genome Atlas (TCGA), was conducted using the single-sample gene set enrichment analysis (ssGSEA) approach. The analysis focused on eight distinct sub-pathways within the DDR process. The results showed a remarkable ability to discriminate between tumor and normal tissues based on the expression patterns of DDR pathway genes [14]. Notably, GC is frequently associated with mutations in key genes involved in homologous recombination, such as BRCA1, BRCA2, partner and localizer of BRCA2 (PALB2), AT-rich interacting domain-containing protein 1A gene (ARID1A), and ataxia telangiectasia mutated (ATM) [15]. As a result, homologous recombination repair (HRR) defects are prevalent in GC, making PARPi an attractive treatment option due to their unique “synthetic lethal” properties.

In this review, we provide a comprehensive summary of the recent advancements and remaining challenges in utilizing PARPi for the treatment of GC. The potential of PARPi in targeting specific vulnerabilities in GC holds promise and could lead to significant improvements in the management of this aggressive cancer.

## 2. Overview of PARP Family Proteins and PARPi

### 2.1. The Structure of PARP Family Members and PARPi

The PARP family is a versatile group of enzymes responsible for protein post-translational modifications and is found in all eukaryotic cells [16]. Through comprehensive investigations, it has been determined that the PARP enzymes comprise at least 17 PARP-related enzymes, including PARP1, PARP2, PARP3, VPARP, Tankyrase-1, and others (Figure 1A) [17,18]. Notably, there is a remarkable sequence similarity within the PARP family, leading to the presence of a highly conserved (ADP-ribosyl) transferase (ART) domain in each protein. This ART domain is responsible for catalyzing the cleavage of nicotinamide adenine dinucleotide (NAD^+^) into nicotinamide and ADP-ribose, enabling ADP-glycosylation modifications on various nucleoproteins [19,20]. Additionally, each PARP enzyme possesses a distinct catalytic domain, governing the diverse mechanisms by which PARP enzymes regulate poly (ADP-ribose) production (Figure 1A) [21,22].

Since the discovery of poly(ADP-ribosyl)ation (PARylation) in 1963, extensive investigations have been conducted on PARP1, a pioneering member of the PARP family, to understand its structure and functionality [23]. Comprising a single peptide chain of 1014 amino acids, PARP1 consists of three distinct regions: the Zn binding domain (ZnI, ZnII, and ZnIII binding domains), the Auto-modification (AMD) domain, and the catalytic (CAT) domain [24]. ZnI and ZnII are responsible for recognizing and binding to DNA damage sites, while ZnIII plays a critical role in the activation process following DNA binding [25,26]. Moreover, PARP1 possesses a WGR domain that functions as both a DNA interactor and sensor for catalytic activity in response to DNA damage (Figure 1B). Interestingly, PARP2 and PARP3 within the PARP family also contain the WGR domain involved in DDR [27,28]. However, PARP2 and PARP3 have a more compact structure compared to PARP1, featuring only a short N-terminal extension from the WGR domain (Figure 1A,B) [16]. Functionally, PARP2 is considered the primary PAR-forming PARP following PARP1 and can compensate for the loss of PARP1 in vivo [29,30]. Hence, the PARPi approved by the US Food and Drug Administration (FDA) (Figure 1C, Table 1) for clinical application primarily targets PARP1 and PARP2, taking advantage of their crucial roles in DNA repair processes.

The design of PARPi primarily revolve around small molecules that contain a nicotinamide/benzamide core. These molecules effectively dock into the NAD^+^ pocket within the ART domain of PARP. By competing with NAD^+^, these inhibitors hinder the catalytic activity of PARP, preventing automatic PARylation and impeding PARP dissociation from DNA (Figure 1B) [24]. As a result, PARP-DNA complexes accumulate, disrupting DNA repair processes, and ultimately inducing cell death [42]. In 2021, Murai et al. made a noteworthy observation that although PARPi possess similar catalytic potency in inhibiting PARP, they exhibit considerable differences (10–40 fold) in PARP capture energy [43]. Further investigations revealed that larger PARPi, such as Olaparib, which binds to the PARP catalytic domain, results in steric hindrance on the HD subdomain that induces conformational rearrangements in the DNA binding domain. This conformational change leads to the formation of a more stable PARP1-DNA complex, enhancing the PARP trapping effect [44]. In contrast, due to its smaller size, Veraparib can solely bind to the catalytic pocket of nicotinamide without engaging with the DNA-binding domain (HD). Consequently, Veraparib exhibits lower levels of PARP capture and cytotoxicity [45]. These findings strongly suggest that, in addition to targeting the catalytic pocket of nicotinamide, the DNA-binding domain of PARP represents a potential site for PARP capture in the development and design of PARPi. This knowledge opens up new avenues for optimizing the design of PARPi to enhance their effectiveness in targeting PARP and improving cancer treatment outcomes.

### 2.2. Evolution from DNA Damage Repair Mechanisms to the Development of PARPi

The impressive tumor-specific cytotoxicity of PARPi are rooted in the pioneering concept of “synthetic lethality” proposed in the preceding century [46,47]. Traditionally, gene A and gene B operate independently in normal physiological conditions. However, while the disruption of either gene alone is tolerable, the simultaneous perturbation of both genes leads to nonviable cells, giving rise to an intriguing interplay between the two genes and resulting in a synthetic lethal interaction (Figure 2A) [48,49].

Genome integrity in healthy cells relies on various DNA repair pathways, including base excision repair for single-strand breaks (SSB), nucleotide excision repair for helical distortions, mismatch repair for replication errors, and a homologous or non-homologous recombination for double-strand break (DSB) repair [50,51,52]. Notably, defects in the homologous recombination repair (HRR) pathway, regulated by BRCA1 and BRCA2 proteins, are linked to multiple tumor types, including breast, ovarian, and gastric cancers [53,54].

Within the HRR pathway, BRCA1 serves as a scaffold protein, and BRCA2 facilitates the recruitment of recombinases such as RAD51 to sites of DNA repair, ensuring efficient and accurate DSB repair (Figure 2B) [55]. Functional mutations in BRCA1 and BRCA2 have been identified in tumor cells, impairing their HRR capacity [56]. Exploiting this vulnerability, PARPi selectively target tumor cells lacking functional BRCA1 or BRCA2 [57].

Upon PARPi administration, DSBs accumulate, while SSBs remain unrepaired. In HRR-deficient tumor cells, the compromised DNA repair machinery resorts to error-prone non-homologous end-joining pathways for DSB repair [58,59]. A further investigation of DNA repair pathways, particularly HRR, and the strategic development of PARPi, offer promising avenues to enhance patient outcomes and broaden therapeutic options in the fight against cancer.

### 2.3. The Application Status of PARPi

A momentous breakthrough in tumor therapy emerged with the introduction of PARPi, profoundly transforming the treatment landscape. Notably, Olaparib, the pioneering PARPi, earned regulatory approval from both the US FDA and the European Medicines Agency (EMA) in 2014. Initially indicated for the maintenance treatment of advanced ovarian cancer with BRCA1/2 mutations, Olaparib showcased unparalleled clinical efficacy [60]. Subsequently, other PARPi, such as Rucaparib, Talazoparib, and Niraparib, obtained regulatory approval, expanding the therapeutic options against various malignancies (Figure 1C, Table 1) [61].

The remarkable effectiveness of PARP extends beyond BRCA-mutated ovarian and breast cancers. Extensive clinical trial data have revealed their therapeutic potential in prostate cancer, pancreatic cancer, and small cell lung cancer (SCLC), regardless of BRCA status [62,63]. This breakthrough discovery has propelled PARPi to the forefront of clinical tumor treatment, establishing it as a paramount first-line therapy.

Nevertheless, the increasing use of PARPi in clinical settings has posed a significant challenge, resulting in the emergence of resistance, both from the beginning and as the treatment progresses [64]. This challenge represents a major obstacle in the field of clinical therapeutic approaches. Since we do not fully understand the underlying mechanisms, the primary strategy revolves around combining different treatment approaches [65]. It is important to note that the cytoprotective phosphatidylinositol 3-kinase (PI3K)-AKT pathway is highly active in most human tumor cells, making it an attractive target for developing combination therapy strategies [66]. Research conducted by Xu J et al. and Zhi W et al., involving ovarian cancer cell lines and animal models, provides empirical evidence that inhibiting the AKT signaling pathway, either through AKT inhibitors or by subjecting cells to nutrient deprivation, significantly enhances the sensitivity of ovarian cancer cells to PARPi. This leads to a synergistic effect in ovarian cancer treatment [67,68].

Furthermore, AKT and the extracellular regulated protein kinase 1/2 (ERK1/2), which are well-known regulators of autophagy, have been implicated in their influence on the expression of DNA repair proteins [69]. Pioneering work by Zai W et al. has unveiled the potential of simultaneously targeting PARP and autophagy, resulting in a surprisingly synergistic effect on the lethality of hepatocellular carcinoma cells (HCC) [70]. This discovery highlights the feasibility of a combined approach involving autophagy and PARPi for effective malignancy treatment. The underlying mechanism for this synergy primarily revolves around the fact that inhibiting autophagy not only negates the cell cycle arrest and checkpoint activation induced by Niraparib but, more importantly, impedes the recruitment of the DNA repair protein RAD51 to double-strand break sites. RAD51, a key enzyme in DNA repair, has emerged as a novel therapeutic target in oncology, with available inhibitors [71]. Researchers have also explored its combination with PARPi, leading to the expected result that the simultaneous administration of RAD51 inhibitors sensitizes breast cancer cells to Olaparib treatment alone [72]. Recent work by Malka MM et al. has introduced a dual inhibitory drug designed to target PARP and a conjugate of RAD51-Olaparib-RAD51 inhibitors [63]. This innovative approach shows promise in overcoming resistance mechanisms observed in breast cancer cells to Olaparib treatment, regardless of their BRCA status. Additionally, similar to RAD51, RECQL5 plays a crucial role in replication, recombination, DNA repair, and transcription [73]. Philip KT et al. have aptly highlighted that even in cancers with proficient HRR, inhibiting RECQL5 can stabilize RAD51, rendering these cancers sensitive to PARPi treatment [74]. These various combinations present new perspectives for expanding the possibilities of clinical tumor therapy involving PARPi.

Recent advancements in small interfering RNA (siRNA) and CRISPR/Cas9 genome editing technologies have propelled the field forward, enabling large-scale and precise screening for synthetic lethal interactions at the cellular level [75,76]. Leveraging these tools, Benjamin H Lok et al. unveiled SLFN11 as a promising predictive biomarker for PARPi sensitivity in SCLC [77]. These breakthrough findings not only provide valuable insights into the complex network of synthetic lethal genes in conjunction with PARP1 but also open up avenues for exploring novel synthetic lethal gene candidates. Consequently, the horizon of PARPi extends far beyond their current applications, offering unprecedented opportunities for their utilization in diverse tumor types [78].

## 3. Impairment of DNA Damage Repair to HRR in GC

### 3.1. DNA Damage Induces Gastric Carcinogenesis

Preserving genome integrity relies on the critical roles of DNA repair mechanisms and DNA damage signaling pathways. In the intricate landscape of cancer, including GC, disruptions in these fundamental processes lead to genomic instability, a common hallmark of malignancies [79]. Epidemiological investigations focusing on GC have revealed the multifaceted nature of this disease, highlighting various factors contributing to its onset. These factors range from dietary patterns and lifestyle choices to genetic predispositions and racial disparities, all converging to promote GC by causing DNA damage (Figure 3A) [80].

*Helicobacter pylori* (*H. pylori*), a microaerophilic gram-negative bacterium, was discovered by Barry Marshall and Robin Warren in 1984. It exhibits remarkable adaptability, colonizing the human stomach [81]. Classified as a class I pathogenic agent for GC by the World Health Organization, *H. pylori* infection holds a prominent position in the global burden of this malignancy [82,83]. With its widespread prevalence, infecting over 50% of the world’s population, the significance of this bacterium in GC development cannot be underestimated [84].

The complex interplay between *H. pylori* and GC encompasses two crucial dimensions. First and foremost, mirroring the pathogenicity witnessed in diverse microbial infections, the inception of chronic inflammation assumes a central role in driving malignant changes. Additionally, *H. pylori* harnesses its virulence determinants, notably the cytotoxin-associated gene A (CagA) and vacuolating cytotoxin A (VacA), to trigger DNA damage and hinder the mechanisms of HRR (Figure 3B) [85]. Within the realm of *H. pylori* infection within the gastric mucosa, whether it be chronic gastritis or intestinal metaplasia, CagA prompts a depletion of BRCA1 in the nucleus. This depletion then triggers the initiation of double-strand DNA breaks and interferes with the functioning of homologous recombination mechanisms. As a result, this series of steps culminates in the destabilization of the genome and expedites the advancement of gastric carcinogenesis [86]. Concurrently, VacA initiates endoplasmic reticulum (ER) stress, disturbs autophagy, and incites apoptosis [76]. Significantly, it is worth highlighting that the generation of reactive oxygen species (ROS) induced by VacA can amass within gastric epithelial cells due to the influence of various virulence factors originating from *H. pylori*. Consequently, this ROS accumulation leads to DNA damage within the host genome, thereby fostering the progression of GC [87].

Apart from *H. pylori*, which is recognized as the primary causative factor, smoking and alcohol consumption have been identified as significant contributors to the development of GC [88,89]. Extensive research has demonstrated that smokers face an approximate 80% higher risk of GC compared to non-smokers [90]. Similarly, individuals with a history of alcohol consumption exhibit a greater susceptibility to GC [91]. Notably, a comprehensive meta-analysis revealed a substantial association between alcohol consumption and GC risk, indicating a 5% increased risk per 10 g/day of alcohol intake [92]. Among individuals with GC who have a history of alcohol abuse, an increased occurrence of elevated TFIIIB-related factor 1 (BRF1) expression and infiltration of MPO-positive cells within tumor tissues was more commonly noted. Furthermore, positive BRCA1 expression in adjacent tissues was observed with greater frequency in these cases [93]. The connection between the consumption of alcohol and the development of GC could be attributed to its capacity to potentially cause DNA damage.

### 3.2. Homologous Recombination Deficiency in GC

A study conducted on a cohort of 10,426 GC patients and 38,153 controls from BioBank Japan investigated the relationship between germline pathogenic variants in 27 cancer susceptibility genes and the risk of GC [94].

In a comprehensive examination of tumor samples collected from a substantial cohort comprising 17,486 patients diagnosed with advanced gastrointestinal cancer, a noteworthy portion (27%) of those with gastric adenocarcinoma displayed deficiencies in their DDR. This was evident through the identification of one or more modifications in DDR-related genes [95]. In the context of the scrutinized DDR genes, the frequencies of genetic alterations were particularly prominent in ARID1A (9.2%) and ATM (4.7%) within this group of patients. Furthermore, although with relatively lower frequencies, modifications were also detected in BRCA2 (2.3%), BRCA1 (1.1%), CHEK2 (1.0%), ATR (0.8%), CDK12 (0.7%), PALB2 (0.6%), CHEK1 (0.1%), and RAD51 (0.1%) [15].

Some cases of gastric adenocarcinomas showcase mutations in ARID1A, which are particularly evident in subtypes linked with microsatellite instability (MSI) and Epstein-Barr virus (EBV), as well as in the less prevalent chromosomal instability (CIN) subtypes [84]. CRISPR/Cas9 gene knockout methodology was utilized by Lo YH et al. to delve into the functional repercussions of removing the ARID1A gene, in addition to silencing TP53 [96]. The outcomes of their study showcased that the deletion of the ARID1A gene led to morphological dysplasia, triggered tumorigenic properties, and prompted mucinous differentiation within gastric organoids. Through a synthetic lethal screening strategy, a connection was unveiled between BIRC5, YM-155, and proliferation when considering the absence of ARID1A [96]. This discovery introduces a fresh therapeutic focal point, tapping into the notion of “synthetic lethality”, as a potential avenue for addressing GC.

Furthermore, the frequencies of ATM and BRCA mutations are notably high in GC patients and have been linked to unfavorable prognostic outcomes [97]. ATM kinase, a nuclear protein, plays a pivotal role in DNA repair signaling and activation of cell cycle checkpoints in response to DNA damage [98]. Experimental evidence suggests that ATM holds promise as a predictive biomarker for the efficacy of PARPi in GC, especially in the context of ATM mutations. The use of PARPi can selectively target P53-deficient GC cells that exhibit sensitivity [99,100]. BRCA, as the pioneering “synthetic lethal” gene targeted by PARPi, exhibits significant differences in risk score, NtAI score, HRD-LOH score, LST score, and HRD score between wild-type and mutant cases of BRCA2 in GC patients, making it a predictive biomarker for the response to DNA-damaging agents in these patients [101,102]. Additionally, BRCA2, along with RAD51, has been identified as a valuable immunohistochemical prognostic marker [103].

Moreover, a study investigating other DDR-associated genes utilized whole exome sequencing (WES) and bioinformatics tools, revealing that PWWP (Pro-Trp-Trp-Pro) domain containing 2B (PWWP2B) is among the frequently mutated genes in tissues of Korean patients with gastric adenocarcinoma [104]. PWWP2B plays a crucial role in DDR and homologous recombination-mediated DSB repair through its interaction with ubiquitin-like with PHD and RING finger domains1 (UHRF1). The mutation status and mRNA expression level of PWWP2B are closely linked to the overall survival of GC patients. Additionally, CHFR (checkpoint protein with FHA and RING finger domains), a tumor suppressor, not only recognizes PARylation signaling at DNA damage sites but also demonstrates that the silencing of CHFR sensitizes GC to PARPi therapy [105].

Overall, an increasing body of evidence supports the frequent occurrence of gene defects in DDR, particularly in the HRR pathway, among GC patients. These defects have a significant impact on prognosis, underscoring the importance of developing targeted therapeutic approaches, such as PARPi, based on the underlying molecular mechanisms.

## 4. Present Status of PARPi in GC Therapy

Building upon the preclinical groundwork mentioned earlier, the use of PARPi shows promising potential in the realm of GC therapeutics. Currently, a significant number of clinical trials involving PARPi, either as standalone treatments or in combination therapies, have either been initiated or are actively underway for the management of GC (Table 2).

### 4.1. Clinical Trials Investigating Monotherapy Using PARPi in GC

As the pioneering PARPi introduced in clinical practice, Olaparib naturally became the initial choice for GC therapy (Figure 1C, Table 1) [106]. Sung Ho Moon et al. conducted a study demonstrating Olaparib’s effective suppression of EBV-positive GC cells in a dose-dependent manner. It induces apoptosis through the EBNA1-ATR-p38 MAPK signaling pathway [107]. However, a disappointing phase III trial investigating Olaparib’s efficacy failed to show a significant improvement in the overall survival among GC patients, leading to its discontinuation without a clear elucidation of the reasons behind this outcome [108].

Nevertheless, utilizing PARPi as monotherapy in GC poses significant challenges. Several key factors contribute to this predicament. Firstly, the complex classification of GC reveals a subset of cases where HRR function remains intact, with no impediments in the HRR pathway. In these instances, DNA damage results from deficiencies in non-homologous end-joining (NHEJ), reducing reliance on the HRR pathway for DDR, thus rendering PARPi ineffective in exerting a “synthetic lethal” effect [109]. Secondly, the well-established drug resistance mechanisms associated with PARPi and the emergence of secondary reverse mutations in HRR proteins minimizes the impact of BRCA mutations on advanced GC. Most GC cases are diagnosed at advanced stages, contributing to the development of drug resistance [110]. Additionally, sequencing studies on GC patients have revealed a high mutation rate in DDR-related proteins, particularly in PARP. Consequently, patients with low levels of PARP expression exhibit lower survival rates, further compromising the efficacy of PARPi-targeted therapy [111]. Moreover, even in cases of GC patients with BRCA1 and BRCA2 gene mutations, resistance to Olaparib treatment persists. RNA analysis has shed light on the absence of transcriptional mutations in BRCA1 and BRCA2 mRNA, despite the presence of gene mutations, which potentially explains the observed resistance to Olaparib among patients with BRCA1/2 mutations [112].

### 4.2. Clinical Trials Evaluating Combination Therapy with PARPi in GC

Due to the limited therapeutic effectiveness of PARPi as a solitary treatment for GC, ongoing research is now primarily focused on exploring the synergistic potential of combining PARPi with other therapeutic agents to manage this malignancy. Building upon the functional attributes of PARP and the underlying principles of PARPi-induced tumor cell death, combination strategies involving DNA damage inducers, AKT inhibitors, angiogenesis inhibitors, and other agents are being investigated as promising approaches for the treatment of GC (Figure 4). These efforts aim to enhance treatment outcomes and provide new avenues for more effective management of this challenging disease (Table 2).

#### 4.2.1. Combining with Inhibitors of DNA Damage Repair

PARPi primarily demonstrate their therapeutic efficacy in tumors with deficiencies in DDR, particularly those with impaired HRR. While PARP1 plays a central role in DDR, other associated proteins also play crucial roles in this intricate process. Leveraging this multifaceted landscape, researchers are favoring the combined use of PARPi with inhibitors targeting these related proteins as an optimal strategy in DDR-targeted cancer treatment.

Checkpoint kinase 1 (Chk1) serves as a critical modulator of cell cycle progression within the DDR. Recent studies have substantiated the profound inhibitory effect of Chk1 inhibitors on HRR-mediated DNA repair. Notably, combining Chk1 inhibitors with PARPi have demonstrated remarkable synergistic anticancer effects, effectively impeding cancer proliferation and inducing apoptosis in GC cells [113]. Furthermore, comprehensive research conducted by Yang Zhao et al. has provided a deeper understanding of how the introduction of a Chk1 inhibitor specifically disrupts homologous recombination-mediated DNA repair in tumor cells that lack inherent resistance to Olaparib. This disruption leads to synthetic lethality in p53-deficient tumors [114]. The primary mechanism behind this effect is attributed to the formation of BRCA1 nuclear foci, which in turn triggers the accumulation of γH2AX, causing DSB, and ultimately resulting in the demise of tumor cells. It is worth noting that while this particular study did not involve GC cell lines, it is significant to recognize that p53 dysfunction is a prevalent occurrence in individuals suffering from GC [113]. This observation also offers valuable insights into the underlying mechanisms that drive the synergistic action of Chk1 inhibitors and PARPi in the context of GC.

Additionally, MUS81, a pivotal endonuclease involved in heterodimer formation with Eme1/Mms4, was identified as a crucial regulator of DDR. Notably, MUS81 knockdown significantly enhances the anticancer effect of Talazoparib [115]. Mechanistically, MUS81 modulates the activation of the ATR/Chk1 pathway, a pivotal signaling cascade in the G2/M checkpoint, which thereby augments the antitumor efficacy of Talazoparib.

Another promising therapeutic candidate is AZD0156, a potent and selective inhibitor of ATM [116]. AZD0156 effectively hinders the repair of DNA damage induced by Olaparib, leading to elevated DNA DSB signaling, cell cycle arrest, and apoptosis. Preclinical studies have demonstrated the enhanced effect of AZD0156 in combination with Olaparib across various lung, gastric, and breast cancer cell lines in vitro, as well as improved efficacy in patient-derived xenograft models of triple-negative breast cancer. Encouragingly, AZD0156 is currently under evaluation in a Phase I clinical trial (NCT02588105) [116].

#### 4.2.2. Combining with AKT Inhibitors

AKT, a serine/threonine kinase, plays a pivotal role as a key component in the PI3K signaling pathway. Upon activation, AKT regulates the activity of numerous downstream proteins involved in critical cellular processes such as survival, proliferation, migration, metabolism, and angiogenesis [117]. Its central role makes AKT dysregulation a common occurrence across various human malignancies [118]. In clinical practice, AKT inhibitors have been utilized for the treatment of different tumor types [119]. Interestingly, AKT activation was detected in approximately 30% of Chinese GC patient tumor biopsies, surpassing the HER2-positive rate (approximately 10–12%) [120]. Currently, a phase II clinical trial is underway to evaluate the efficacy and safety of the AKT kinase inhibitor MK2206 as a second-line treatment option for GC patients [121]. Unfortunately, the trial yielded unfavorable outcomes, indicating that monotherapy with MK2206 does not confer a survival advantage in GC patients.

The combination of PI3K and PARPi emerged as a promising therapeutic strategy in breast, ovarian, and prostate cancer. Recent investigations highlighted the inhibitory effects of PARPi on the PI3K/AKT pathway in GC cells. Notably, the upregulation of the ClC-3/SGK1 axis has been implicated in enhancing the PARPi-induced inhibition of the PI3K/AKT pathway, promoting effective tumor suppression [122]. In a study conducted by Lin Yang et al., the therapeutic efficacy of a combined treatment with the PI3K inhibitor BKM120 and Olaparib was evaluated in GC cells [123]. Remarkably, the combined therapy demonstrated a significant effectiveness in suppressing the proliferation, invasion, and migration of GC cells, as evidenced by MTS and clonogenicity assays. These findings underscore the potential of combining PI3K inhibition and PARPi as a promising treatment strategy for GC.

#### 4.2.3. Linked to Antiangiogenic Treatment

Vascular endothelial growth factor (VEGF) plays a crucial role in promoting angio-genesis and tissue regeneration [124]. Inhibitors targeting VEGF receptors (VEGFR) have demonstrated effective inhibition of tumor angiogenesis, thereby limiting nutrient acquisition by tumor cells and ultimately achieving therapeutic outcomes [125]. TC24, an inhibitor of histone deacetylase (HDAC), exhibits therapeutic potential by targeting tumor angiogenesis and restraining the proliferation of GC cells through the downregulation of hypoxia-inducible factor-1α (HIF-1α) and VEGF [126].

Emerging evidence indicates that the inhibition of vascular endothelial growth factor receptor 3 (VEGFR3) leads to reduced levels of BRCA1/2 and impedes cell growth in vitro. Notably, VEGFR3 inhibitors were shown to induce the downregulation of HRR-related proteins, including BRCA1/2 and RAD51, which holds promise in augmenting the sensitivity of PARPi [127]. Moreover, it has been observed that the immune pathways are activated, and the programmed death-ligand 1 (PD-L1) expression is up-regulated on tumor cells as a consequence of the unrepaired DNA damage resulting from PARPi treatment. This phenomenon, in turn, enhances the sensitivity to immune checkpoint inhibitors (ICIs) [128]. Currently, Phase II clinical trials investigating the combined administration of ICIs and PARPi, with or without VEGFR inhibitors or conventional chemotherapy (CT), are underway, aiming to explore the synergistic potential of these therapeutic approaches [128].

#### 4.2.4. Administered with Chemotherapeutic Agents

Interestingly, besides their primary role as therapeutic agents, PARPi demonstrated their potential as sensitizers in GC treatment. Resistance to various chemotherapeutic drugs is a common challenge encountered in GC patients. Huafu Li et al. con-ducted a sequence analysis of oxaliplatin-resistant and non-oxaliplatin-resistant organoids, revealing the significant involvement of PARP as a crucial gene mediating oxaliplatin (OXA) resistance. The inhibition of the base excision repair pathway by PARP was identified as the underlying mechanism driving OXA resistance, with the expression level of PARP showing a significant correlation with OXA resistance. Intriguingly, the combined use of PARPi, specifically Olaparib, and OXA demonstrated a remarkable inhibition of growth and viability in OXA-resistant organoids, both in vivo and in vitro [129]. Moreover, Qiang Wang et al. observed a substantial increase in PARP activity in cisplatin-treated GC-resistant patients. The application of PARPi effectively impeded DNA-dependent protein kinase catalytic subunit (DNA-PKcs) stability, consequently impairing the repair capacity of DNA DSB through the NHEJ pathway. This augmentation of DNA damage and apoptosis in cisplatin-induced drug-resistant GC cells provide evidence of the potential of PARPi as an adjuvant drug to enhance the efficacy of chemotherapy agents [130]. Collectively, these findings suggest that PARPi hold promise as a sensitizer in combination with chemotherapy drugs, and serve as an adjunct therapy to improve treatment outcomes in GC.

#### 4.2.5. Other Combinations

A recent study revealed an intriguing association between PARPi and photodynamic therapy (PDT). PDT, which utilizes photosensitizers and light-induced photo-chemical reactions in target tissues, demonstrated the therapeutic potential in various cancer types [131]. The initial-generation photosensitizer obtained approval from the Japanese National Health Insurance for treating different cancers, including GC [132,133]. Furthermore, the second-generation photosensitizer Talaporfin showed promise, not only in the radical treatment of early GC, but also as a safe, effective, and non-invasive therapeutic approach for elderly patients with advanced GC [134]. Consequently, there is great anticipation regarding the synergistic outcomes anticipated from the combination of PDT and Talaporfin in GC treatment.

Mamoru Tanaka and colleagues conducted a study to investigate the effects of PDT in combination with Olaparib on GC cells. Specifically, they incubated the MKN45 GC cell line with Talaporfin in the presence or absence of Olaparib, followed by irradiation [135]. The results showed that Talaporfin PDT combined with Olaparib led to the increased accumulation of PARP1 in chromatin, which is indicative of the entrapment of PARP1-DNA complexes. This combination therapy demonstrated notable tumor-killing effects both in vitro and in vivo, with the efficacy being dependent on the concentration of the treatment. These findings highlight the potential of Talaporfin PDT in combination with Olaparib as a promising therapeutic approach for GC.

## 5. Conclusions and Future Prospects

GC continues to pose a significant global health challenge, urging the exploration of novel therapeutic approaches. Recent extensive investigations into the intricate inter-play between PARP and DDR pathways have illuminated the significance of HRR deficiency and the potential of PARPi to exploit synthetic lethality in GC, making PARP a promising therapeutic target in the field.

Multiple clinical trials have evaluated the efficacy of PARPi as a solitary treatment, but these have unveiled certain limitations. These limitations stem from the intricate nature of GC’s pathogenesis and progression, as well as the significant inter-individual variability among patients. Notably, some patients maintain intact homologous recombination mechanisms, impacting the clinical applicability of PARPi. As a result, the inclusion of mRNA expression analysis for homologous recombination genes in patient evaluations becomes crucial to enable precise treatment decisions. Within clinical settings, the combination approach has demonstrated enhanced effectiveness when compared to single-agent interventions. Importantly, the promising potential of combining PARPi with other drugs was showcased in the management of GC. This effect is largely attributed to the diverse roles that PARP1 plays across various physiological processes. Recent research emphasized the role of PARP1 as a pro-inflammatory enzyme, contributing to the modulation of immune and inflammatory pathways. Consequently, the utilization of PARPi can activate inflammatory signaling pathways such as cGAS-STING and NFκB [136,137]. Furthermore, emerging data hint at potential ramifications for tumor immunity, holding promise for achieving therapeutic goals by manipulating these pathways through inhibitory or activating tactics.

In conclusion, PARPi advances towards clinical application, highlighting the need for further research to optimize treatment strategies and develop personalized approaches, and ultimately improving therapeutic outcomes for patients with GC. Despite these challenges, PARPi represents a promising new treatment option for patients with GC. Ongoing research should be focused on investigating how to improve the efficacy and safety of PARPi in this patient population.

## Figures and Tables

**Figure 1 cancers-15-05114-f001:**
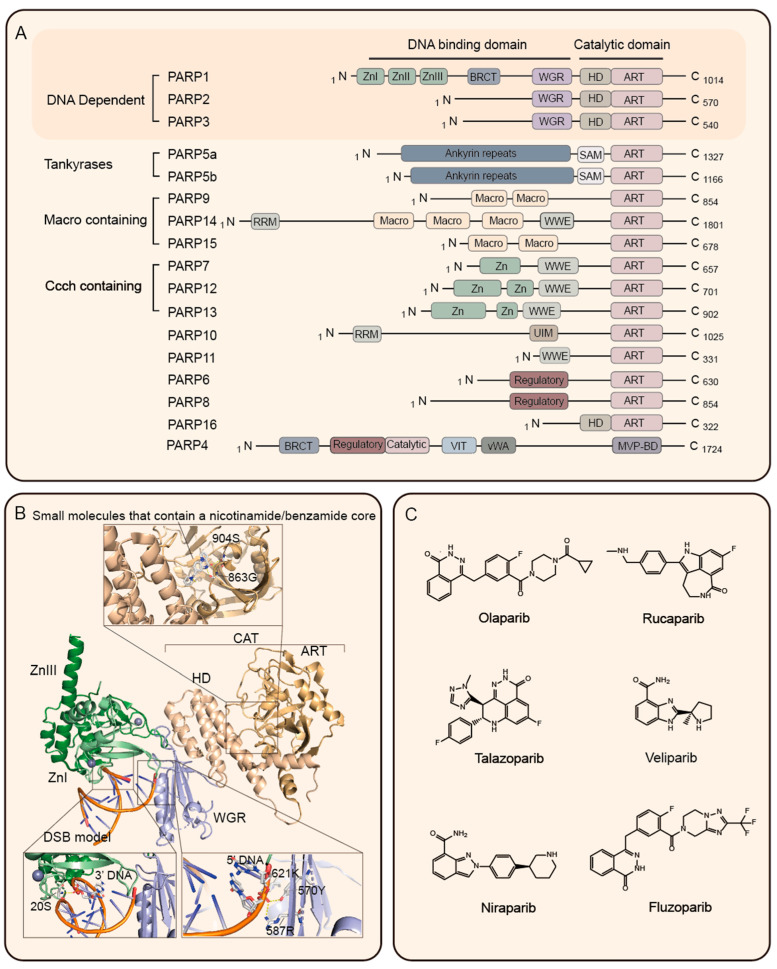
PARP family and PARPi. (**A**) Schematic diagram of domain architecture of PARP family members; (**B**) schematic of the domains of PARP1 (upper panel); crystal structure of PARP1 bound to double-strand DNA (PDB: 4DQY) (lower panel); (**C**) the chemical structures of FDA-approved PARPi. Abbreviations: CCCH, Cys-Cys-Cys-His; BRCT, breast cancer susceptibility gene C terminus; ZnI, ZnII, ZnIII, and Zinc finger; WGR, Tryptophan—Glycine–Arginine; HD, helical domain; macro, macrodomain; ART, Ankyrin repeat domains; RRM, RNA recognition motif; UIM, ubiquitin interaction motif; VIT, Vault protein Inter-Alpha-Trysin; WWE, Tryptophan–Tryptophan–Glutamiac; vWA, von Willebrand type A; MVP-BD, Major-vault particle binding domain.

**Figure 2 cancers-15-05114-f002:**
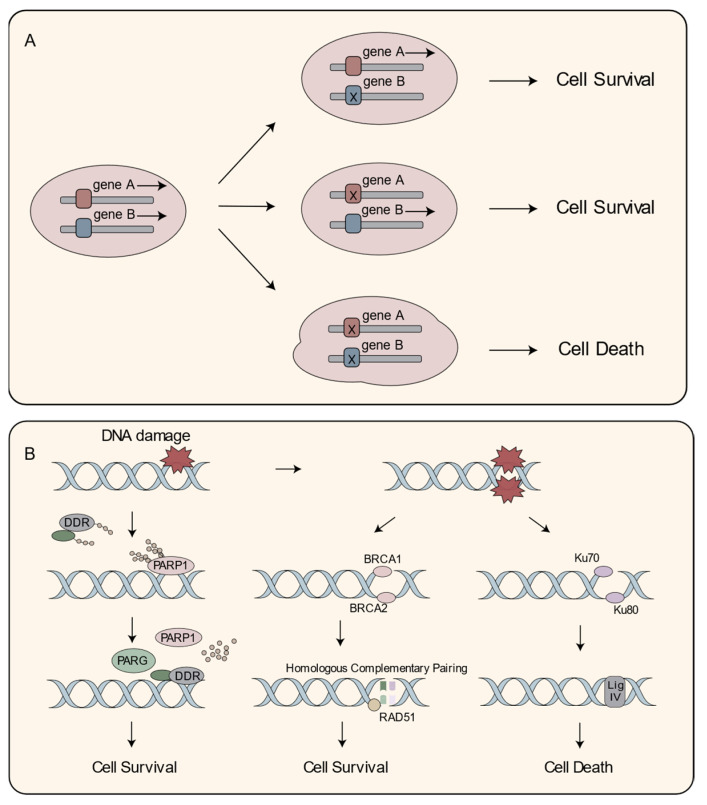
The concept of synthetic lethality and its implementation in therapeutics’ applications of PARPi. (**A**) The principle of Synthetic lethality. In a cell, the mutation or inactivation of either of two genes does not lead to cell death; however, the concurrent mutation or failure of both genes results in cell death; (**B**) PARP1 functions as a detector of SSB and attracts proteins that aid in DDR. The mode of action for PARPi involves synthetic lethality in cells that are defective in homologous recombination HRR. BRCA1 serves as a foundational support protein, while BRCA2 assists in the recruitment of recombinases such as RAD51 to sites requiring DNA repair, thereby ensuring precise and efficient repair of DDR. By employing the concept of synthetic lethality, PARPi has the capacity to specifically eliminate tumor cells harboring mutations in BRCA1/2, all the while sparing normal cells from harm. Abbreviations: PARP, poly (adenosine diphosphate-ribose) polymerase; PARG, poly (adenosine diphosphate-ribose) polymerase; BRAC, breast cancer susceptibility gene; DDR, DNA-damage response; SSB, single-strand break; DSB, double-strand break; HRR, homologous recombination DNA repair; RAD51, RAD51 Recombinase; LigIV, DNA ligase IV.

**Figure 3 cancers-15-05114-f003:**
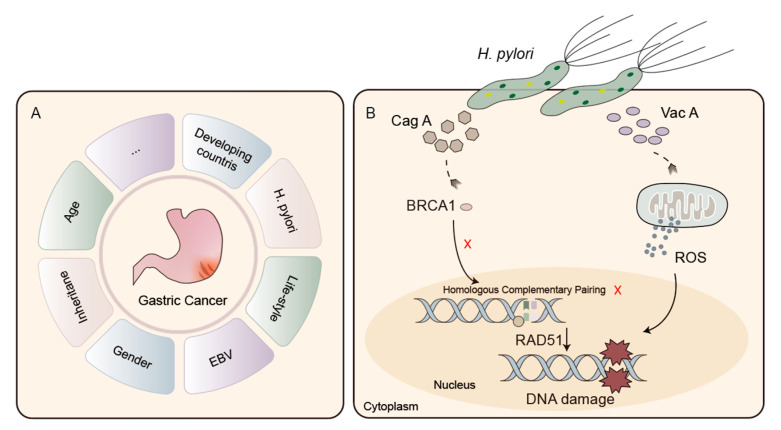
Known risk factors for GC. (**A**) GC is influenced by factors such as dietary habits, lifestyle, genetic susceptibilities, and variations across different racial groups, all of which collectively contribute to the induction of DNA damage. (**B**) *H. pylori* hinders the process of HRR. The CagA-driven reduction of BRCA1 within the nucleus stimulates the initiation of double-strand DNA breaks and disrupts the homologous recombination process. Furthermore, the production of by VacA can amass within gastric epithelial cells, inducing DNA damage in the host genome and thus fostering the advancement of GC. Abbreviations: ROS, reactive oxygen species; GC, gastric cancer; *H. pylori*, Helicobacter pylori; CagA; VacA, cytotoxin-associated gene A vacuolating cytotoxin A; EBV, Epstein-Barr virus; BRAC, breast cancer susceptibility gene; RAD51, RAD51 Recombinase; HRR, homologous recombination repair.

**Figure 4 cancers-15-05114-f004:**
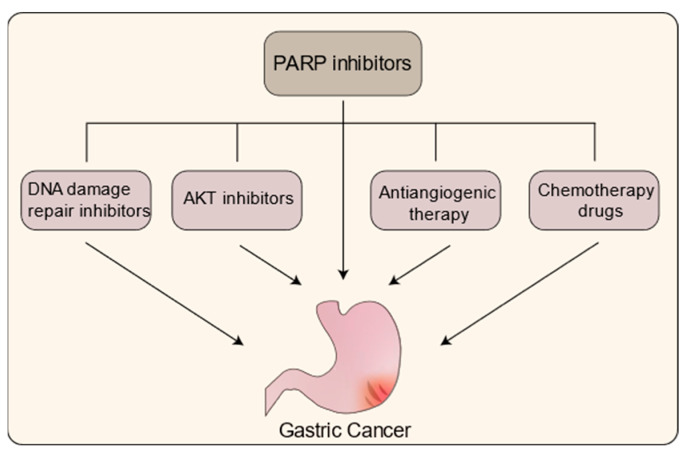
PARPi are utilized in both monotherapy and in conjunction with other medications to manage GC.

**Table 1 cancers-15-05114-t001:** Evolution of the indications for approved PARPi.

PARPi	Year of Approval	Company	Indication	Refs
Olaparib	2014	AstraZeneca; MSD	Ovarian cancer	[31]
			Breast cancer	[32]
			Pancreatic cancer	[33]
			Prostate cancer	[34]
Rucaparib	2016	Clovis Oncology	Ovarian cancer	[35]
			Breast cancer	[36]
			Pancreatic cancer	[37]
Niraparib	2017	Tesaro	Ovarian cancer	[38]
			Breast cancer	[39]
Talazoparib	2018	Pfizer	Breast cancer	[40]
Fluzoparib	2020	Hengrui Medicine	Ovarian cancer	[41]

**Table 2 cancers-15-05114-t002:** Trials Assessing PARPi Alone/in Combination in GC.

PARPi	Combination	Registrational Clinical Trail (s)	Indication	Identifier
Olaparib	EP0057	Phase II	Gastric Cancer	NCT05411679
Olaparib	MEDI4736, Bevacizumab	Phase I/II	Gastric Cancers	NCT02734004
Olaparib	Pembrolizuma, Paclitaxel	Phase I/II	Gastric Cancer Stage IV	NCT04592211
Fluzoparib	Apatinib, Paclitaxel	Phase I	Recurrent and Metastatic Gastric Cancer	NCT03026881
Pamiparib	/	Phase II	Advanced or Inoperable Gastric Cancer	NCT03427814
Olaparib	Paclitaxel, Placebo	Phase II	Gastric Cancer	NCT01063517
Veliparib	Carboplatin, Paclitaxe, Folfiri	Phase I	Gastric Cancer	NCT02033551
Olaparib	Ramucirumab	Phase I/II	Metastatic Esophageal Carcinoma, Metastatic Gastric Carcinoma, Metastatic Gastroesophageal Junction Adenocarcinoma, Recurrent Esophageal Carcinoma, Recurrent Gastric Carcinoma, Recurrent Gastroesophageal Junction Adenocarcinoma, Stage III Esophageal Cancer AJCC v7, Stage IV Esophageal Cancer AJCC v7, Stage IV Gastric Cancer AJCC v7, Unresectable Esophageal Carcinoma, Unresectable Gastric Carcinoma, Unresectable Gastroesophageal Junction Adenocarcinoma	NCT03008278
Veliparib	/	Phase I	Gastric Cancer	NCT01123876
Olaparib	Paclitaxel, Pembrolizumab	Phase II	Advanced Gastric Adenocarcinoma	NCT04209686

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
