# Peer review of "Advancements and Obstacles of PARP Inhibitors in Gastric Cancer"

_cancers, 2023, doi:10.3390/cancers15215114_

Round 1

Reviewer 1 Report

Chen et al’s review is entitled as “Advancements and Obstacles of PARP Inhibitors in Gastric Cancer”. The review is quite well structured and written, taking in account of recent relevant publications in the field. The Review provides a comprehensive summary of the recent advancements and the challenges for utilizing PARPi for the treatment of GC. Considering the fact that GC harbours many DDR mutations, especially HRD, it is important to understand the DDR defective pathogenesis of GC and devopment of effective therapy, where PARPi has remarkable promise. I have following few suggestions for further improving the review: 

1. Line 65-69: The whole review is based on the premises that GC are associated with HRD due to mutations in many HR regulating genes. Instead of citing a review [15] for this, it is better to cite original articles which show the mutations of HR genes in GC. 

2. Line 174: Typos – DDB

3. Line 189: “The remarkable effectiveness of PARP extends beyond BRCA-mutated ovarian and 189 breast cancers.”. This is an emerging concept for extending therapeutic benefits to HR-proficient breast and ovarian cancer patients. Authors should expand this part by adding a dedicated paragraph with recent citations for combination treatment approach with PARPi and other agents eg., (RECQL5, RAD51, autophagy, AKT etc), especially in breast and ovarian cancers. Understanding these recent concepts may also help using PARPi combination therapy for GC. 

4.  Line 366-368: the citation 104 may not be correct refence to show the effect of Olaparib and CHK1i. I found the study is mostly related to AsPC-1 (Pancreatic cancer) and H1299 (lung) cells. Author should revisit the reference and revise it.

Author Response

Dear Reviewers and Editors,

We would like to thank the reviewer for providing thoughtful, constructive and positive comments on our manuscript (cancers-2642879). We thank the reviewer for the supportive remark that “The review is quite well structured and written, taking in account of recent relevant publications in the field. The Review provides a comprehensive summary of the recent advancements and the challenges for utilizing PARPi for the treatment of GC. Considering the fact that GC harbours many DDR mutations, especially HRD, it is important to understand the DDR defective pathogenesis of GC and devopment of effective therapy, where PARPi has remarkable promise.”

A point-by-point response to each comment is listed below.  All modifications made in the text were shown in blue. Please see the attachment.

Comments 1: [Line 65-69: The whole review is based on the premises that GC are associated with HRD due to mutations in many HR regulating genes. Instead of citing a review [15] for this, it is better to cite original articles which show the mutations of HR genes in GC.]

Response 1: Thank you for pointing this out. we have reevaluated the relevant article that elucidates genetic mutations associated with the homologous recombination repair (HRR) pathway in GC and have accordingly updated our references. Subsequently, we have incorporated these revisions into the manuscript (Line 64 to 67). The literature we recite is “Fan Y, Ying H, Wu X, Chen H, Hu Y, Zhang H, Wu L, Yang Y, Mao B, Zheng L. The mutational pattern of homologous re-combination (HR)-associated genes and its relevance to the immunotherapeutic response in gastric cancer. Cancer Biol Med. 2020 Nov 15;17(4):1002-1013. doi: 10.20892/j.issn.2095-3941.2020.0089” (Line 574 to 576).

Comments 2: [Line 174: Typos – DDB]

Response 2: We appreciate your observation, and we regret any inadvertent spelling errors that may have arisen due to our oversight. We have rectified the "DDB" to "DDR" on line 167 of the manuscript.

Comments 3: [Line 189: “The remarkable effectiveness of PARP extends beyond BRCA-mutated ovarian and 189 breast cancers.”. This is an emerging concept for extending therapeutic benefits to HR-proficient breast and ovarian cancer patients. Authors should expand this part by adding a dedicated paragraph with recent citations for combination treatment approach with PARPi and other agents eg., (RECQL5, RAD51, autophagy, AKT etc), especially in breast and ovarian cancers. Understanding these recent concepts may also help using PARPi combination therapy for GC.]

Response 3: We appreciate your attention to this matter and fully endorse your suggestion. Your valuable recommendations contribute to the richness and thoroughness of our articles, and we are pleased to incorporate this section. We have integrated the following components into lines 187 to 221 of the manuscript. If you identify any areas that may require improvement, we eagerly invite your feedback and guidance for necessary adjustments.

[ Nevertheless, the increasing use of PARP inhibitors in clinical settings has posed a significant challenge, resulting in the emergence of resistance, both from the beginning and as treatment progresse. This challenge represents a major obstacle in the field of clinical therapeutic approaches. Since we have not fully understood the underlying mechanisms, the primary strategy revolves around combining different treatment approaches. It is important to note that the cytoprotective phosphatidylinositol 3-kinase (PI3K)-AKT pathway is highly active in most human tumor cells, making it an attractive target for developing combination therapy strategies. Research conducted by Xu J et al. and Zhi W et al., involving ovarian cancer cell lines and animal models, provides empirical evidence that inhibiting the AKT signaling pathway, either through AKT inhibitors or by subjecting cells to nutrient deprivation, significantly enhances the sensitivity of ovarian cancer cells to PARP inhibitors, leading to a synergistic effect in ovarian cancer treatment.

Furthermore, AKT and extracellular regulated protein kinase 1/2 (ERK1/2), which are well-known regulators of autophagy, have been implicated in their influence on the expression of DNA repair proteins. Pioneering work by Zai W et al. has unveiled the potential of simultaneously targeting PARP and autophagy, resulting in a surprisingly synergistic effect on the lethality of hepatocellular carcinoma cells (HCC). This discovery highlights the feasibility of a combined approach involving autophagy and PARP inhibitors for effective malignancy treatment. The underlying mechanism for this synergy primarily revolves around the fact that inhibiting autophagy not only negates the cell cycle arrest and checkpoint activation induced by Niraparib but, more importantly, impedes the recruitment of the DNA repair protein RAD51 to double-strand break sites. RAD51, a key enzyme in DNA repair, has emerged as a novel therapeutic target in oncology, with available inhibitors. Researchers have also explored its combination with PARP inhibitors, leading to the expected result that simultaneous administration of RAD51 inhibitors sensitizes breast cancer cells to Olaparib treatment alone. Recent work by Malka MM et al. has introduced a dual inhibitory drug designed to target PARP and a conjugate of RAD51-Olaparib-RAD51 inhibitors. This innovative approach shows promise in overcoming resistance mechanisms observed in breast cancer cells to Olaparib treatment, regardless of their BRCA status. Additionally, similar to RAD51, RECQL5 plays a crucial role in replication, recombination, DNA repair, and transcription. Philip KT et al. have aptly highlighted that even in cancers with proficient homologous recombination (HR), inhibiting RECQL5 can stabilize RAD51, rendering these cancers sensitive to PARP inhibitor treatment. These various combinations present new perspectives for expanding the possibilities of clinical tumor therapy involving PARP inhibitors.]

Comments 4: [Line 366-368: the citation 104 may not be correct refence to show the effect of Olaparib and CHK1i. I found the study is mostly related to AsPC-1 (Pancreatic cancer) and H1299 (lung) cells. Author should revisit the reference and revise it.

Response 4: We would like to express our gratitude to the reviewer for their valuable suggestion. After a thorough examination of the relevant literature, we have realized that the article in question did not conduct experimental investigations involving gastric cancer cells. We deeply regret this unintentional misquotation. It's important to note, however, that the dysfunction of p53 is a commonly observed phenomenon in individuals suffering from gastric cancer. This observation also offers significant insights into the underlying mechanisms responsible for the synergistic effects of Chk1 inhibitors and PARP inhibitors in the context of gastric cancer. In response to this issue, we have made revisions and provided clarification for this matter in lines 393 to 403 of the manuscript.

[Furthermore, comprehensive research conducted by Yang Zhao et al. has provided a deeper understanding of how the introduction of a Chk1 inhibitor specifically disrupts homologous recombination-mediated DNA repair in tumor cells that lack inherent resistance to Olaparib. This disruption leads to synthetic lethality in p53-deficient tumors. The primary mechanism behind this effect is attributed to the formation of BRCA1 nuclear foci, which in turn triggers the accumulation of γH2AX, causing DNA double-strand breaks (DSB), ultimately resulting in the demise of tumor cells. It is worth noting that while this particular study did not involve gastric cancer (GC) cell lines, it is significant to recognize that p53 dysfunction is a prevalent occurrence in individuals suffering from GC. This observation also offers valuable insights into the underlying mechanisms that drive the synergistic action of Chk1 inhibitors and PARP inhibitors in the context of GC.]

Reviewer 2 Report

This review addresses a timely issue about the use of PARP inhibitors in gastic cancer and summarizes very adequately the importance of DDR in designing new therapeutic strategies in GC. Globally the review is wellorganised and pinpoint the different aspects of GC therapy related to PARP inhibition in a bold and complete way

Here are the major changes that need to be done:

1. an effort should be made by the authors in reducing the paragraphs concerning the general aspects of PARP family of proteins as well a asthe concept of synthetic lethality as this topics have been addressed repeatedly in a large number of reviews

2. Additionally, the quality of table 2 needs to be improved as the current organisation looks untidy and lacks a a number of identifiers

3. References 125 and 126 are the same. A new reference (126) needs to be included

1. Figure 1: spelling of "DNA-dipendent " is not correct

2. Veriparib should be corrected for veliparib

3. Line 142: references 47 and 48 are of the "current" century and not of the preceding century

Author Response

Dear Reviewers and Editors,

We are grateful for the opportunity provided by you and the reviewers to publish our paper titled "Advancements and Challenges of PARP Inhibitors in Gastric Cancer" in Cancers as an invited review article for the special issue titled "Advances in Cancer Therapeutics." Furthermore, we are thrilled to receive the reviewer's positive feedback, describing this article as follows: "Globally the review is wellorganised and pinpoint the different aspects of GC therapy related to PARP inhibition in a bold and complete way."

A point-by-point response to each comment is listed below.  All modifications made in the text were shown in blue.

Comments 1: [an effort should be made by the authors in reducing the paragraphs concerning the general aspects of PARP family of proteins as well as the concept of synthetic lethality as this topic have been addressed repeatedly in a large number of reviews]

Response 1: We express our gratitude to the reviewer for their insightful critique and valuable suggestions. We have promptly removed the specific section from the manuscript in response to your feedback. Nevertheless, in order to maintain the article's overall integrity and comprehensiveness, we have thoughtfully retained some relevant content. We kindly seek your understanding regarding this decision. These corrections have been implemented in the revised manuscript (Line 76 to 103).

Comments 2: [Additionally, the quality of table 2 needs to be improved as the current organization looks untidy and lacks a number of identifiers]

Response 2: We appreciate your observation, and in response, we have thoroughly revised Table 2 to align with your expectations and to ensure its overall quality. These corrections have been incorporated into the revised manuscript(Line 370-371). 

Comments 3: [References 125 and 126 are the same. A new reference (126) needs to be included]

Response 3: We have successfully updated the citation from No. 126 to its correct reference in the revised manuscript, which is now correctly documented as No. 138.

Response to Comments on the Quality of English Language

Point 1: Figure 1: spelling of "DNA-dipendent " is not correct

Response 1: We have corrected this in the revised manuscript (Line 134).

Point 2: Veriparib should be corrected for veliparib

Response 2: We have corrected this in the revised manuscript (Line 134).

Point 3: Line 142: references 47 and 48 are of the "current" century and not of the preceding century

Response 3: We are grateful for your keen observation and for bringing the issue to our attention. In the revised manuscript (Lines 126 to 127), we have properly cited references NO.47 and NO.48, both of which were published in the previous century.
